# HoVer: Holistic Verification for Semantic-Aware Speculative Generation

## Abstract

We introduce *HoVer*, a semantic-aware speculative generation framework that accelerates large language model (LLM) inference without retraining. *HoVer* employs Holistic Verification: a lightweight draft model generates a complete candidate output, and a larger base model then *verifies* it holistically and, if necessary, *revises* from the first detected error. Unlike token-level speculative decoding, which enforces distributional consistency one token at a time, *HoVer* operates at the *semantic* level, amortizing verification cost over longer spans of text. At the core of our design is a prefill-only, single-pass *prefix verification* mechanism that uses a custom attention mask to identify the earliest error across multiple prefixes simultaneously. This makes verification compute-bound with negligible KV-cache overhead and enables continuation from the last safe prefix instead of regenerating from scratch. Across model families, *HoVer* achieves $\sim$1.2$\times$–3.1$\times$ latency reduction with minimal accuracy loss across general and math benchmarks. The approach is orthogonal to token-level speculation and can be combined with it for further gains.

## 1 Introduction

As large language models (LLMs) scale, their inference cost increasingly dominates deployment budgets. Reducing latency and server load without compromising quality has therefore become a central research goal (Kwon et al., 2023; Yu et al., 2022; Zhong et al., 2024). Among existing techniques, *speculative generation* (Leviathan et al., 2023; Chen et al., 2023) is especially attractive: a lightweight draft model proposes tokens that the base model then verifies. Classical speculative *decoding* is *lossless*: through rejection sampling, the final output distribution exactly matches that of the base model, so any accepted tokens are guaranteed to be distributionally consistent.

While token-level speculation provides rigorous correctness, it constrains acceleration to per-token agreement. Many user-facing tasks admit a coarser but still reliable unit of verification: *semantic spans*. In practice, correctness is often determined by whether a prefix remains on a correct solution trajectory (e.g., problem setup, derivation scaffolding, or factual context), not by matching the base model's probability at each token position. This observation opens a complementary axis of speedup: amortize verification over longer semantic units, reuse any prefix that is semantically sound, and only regenerate from the first incorrect point onward.

Prior works have also recognized the potential of semantic-level speculation. In particular, SpecReason (Pan et al., 2025) and Speculative Thinking (Yang et al., 2025b) apply semantic verification to reasoning-oriented tasks by checking intermediate chain-of-thought steps. However, these methods are tailored to reasoning traces only and rely on ambiguous step boundaries, limiting their applicability. By contrast, our approach is designed for *general-purpose text generation*: *HoVer* verifies arbitrary spans without depending on CoT segmentation, making it broadly applicable beyond reasoning tasks.

Motivated by the potential of *semantic-level verification* as a complementary axis of acceleration, we present *HoVer*, a speculative generation framework based on holistic verification: a small model drafts a complete candidate output, and a larger base model then *verifies* it and, if necessary, *revises* from the first detected error. Two design choices make this efficient and broadly applicable. First, instead of stepwise checks, *HoVer* performs a single *prefill-only*, compute-bound holistic verification pass that simultaneously judges multiple prefixes via a custom attention mask—avoiding sequential

Figure 1: Comparison of *Speculative Decoding* and our *HoVer* framework. Speculative Decoding verifies tokens sequentially over multiple passes, while *HoVer* checks all prefixes in one prefill and resumes from the last correct prefix, making it more flexible and efficient.

verification and preventing KV-cache growth. Second, by operating on flexible spans rather than predefined reasoning steps, *HoVer* applies naturally to both reasoning and open-domain text generation. *HoVer* is orthogonal to token-level speculative decoding: the latter preserves losslessness at the token level, while our semantic verification adds a new axis of acceleration; the two can be combined for compounded gains.

We make the following contributions:

- **Semantic-level speculation for general text generation.** We formalize and exploit verification at the level of semantic spans, enabling reuse of correct prefixes rather than enforcing token-by-token agreement. Unlike prior works limited to reasoning tasks, *HoVer* applies broadly across domains.

- **Parallel, prefill-only prefix verification.** We introduce a custom attention mask that lets the base model judge many prefixes in a *single* forward pass, keeping verification efficient. Upon detecting the earliest error, the base model continues from the last safe prefix instead of regenerating from scratch, reducing redundant decoding.

- **Compatibility with lossless token-level speculation.** Our approach is orthogonal to classical speculative decoding (Leviathan et al., 2023; Chen et al., 2023) and can be layered with it for additional speedups.

Empirically, when using Qwen3-235B as the base model, *HoVer* achieves $\sim 1.2\times$–$3.1\times$ latency reductions with negligible accuracy loss across general and math benchmarks, demonstrating that semantic-level verification is a practically applicable path to faster LLM serving.

## 2 RELATED WORKS

### 2.1 SPECULATIVE DECODING

Speculative decoding has recently seen several enhancements to draft tree structures, feature-level modeling, and single-model drafting. The EAGLE series (EAGLE-1, EAGLE-2) (Li et al., 2024a;b) introduces static then dynamic draft trees, enabling context-aware token drafting with lossless speedups of 3–4.3$\times$. Dynamic Depth Decoding (Brown et al., 2024) further improves on EAGLE-2 by adapting draft depth per context, gaining additional speed without accuracy loss. EAGLE-3 (Li et al., 2025) relaxes the feature-prediction constraint by fusing multiple semantic feature levels during training and inference, achieving better speedups. Alternatives like Medusa (Cai et al., 2024) avoid separate small models by attaching lightweight "draft heads" directly to the base model.

These variants of speculative decoding still rely on *token-level* distributional alignment, where the base model sequentially verifies agreement on each token. In contrast, *HoVer* shifts the focus to *semantic-level* verification: instead of matching token probabilities, the base model evaluates whether draft prefixes remain semantically correct. This is done in a single prefill pass, after which

the base model continues from the last verified prefix. Figure 1 illustrates this contrast between token-level speculative decoding and our approach.

## 2.2 SEMANTIC-LEVEL SPECULATIVE DECODING

We are not the first to consider semantic-level speculation. Prior works such as JudgeDecoding (Bachmann et al., 2025) enables a larger acceptance length (about 19 tokens on average) by training a small network to recognize semantic-level alignment based on the last layer of network features. SpecReason (Pan et al., 2025), Lookahead Reasoning (Fu et al., 2025) and Speculative Thinking (Yang et al., 2025b) apply semantic verification to reasoning steps in a chain-of-thought (CoT), enabling speed-up in the reasoning process. In these approaches, a small draft model proposes an intermediate reasoning step, which the larger model then accepts or rejects before the next step is produced.

Our work is different from prior works in semantic level alignment in the following way: (1) Compared to JudgeDecoding (Bachmann et al., 2025), HoVer is training-free, and focuses on the holistic semantic alignment over the whole answer/specific prefix spans, while JudgeDecoding is an extension of speculative decoding by increasing the acceptance length. (2) Compared to SpecReason (Pan et al., 2025) and Lookahead Reasoning (Fu et al., 2025), Hover has a different application domain. These methods only target reasoning traces, whereas our work focuses on *general text generation*, making semantic-level speculation applicable even outside of explicit CoT settings. Further, verification in prior works is inherently *sequential*, since each step must be checked before moving forward. By contrast, our method performs *holistic*, parallel prefix verification in a single prefill-only pass.

## 2.3 MODEL CASCADES

Cascaded inference, as an alternative to speculative sampling, reduces large-model cost by using a sequence of increasingly powerful models and deferring to larger models only when needed. Methods differ by their deferral criterion: confidence-threshold cascades (Lebovitz et al., 2023) and agreement-based cascades (Kolawole et al., 2025) use per-tier confidence or ensemble agreement to decide whether to escalate. Recent work unifies and formalizes these ideas under a single model-selection framework, showing when routing, cascading, or a hybrid "cascade-routing" strategy is optimal (Dekoninck et al., 2025). At the system level, CASCADIA (Jiang et al., 2025) addresses large-scale serving of cascaded models.

While cascaded inference also exploits the trade-off between model size and accuracy, *HoVer* differs in two key ways. First, instead of discarding the smaller model's output when escalation occurs, we reuse the draft and let the larger model regenerate only from the position of the first detected error. Second, rather than requiring a separate deferral mechanism, we employ the base model itself as the verifier, simplifying deployment and reducing reliance on additional models.

# 3 HOVER DESIGN

## 3.1 INFERENCE FRAMEWORK

We design and implement a multi-stage framework that alternates between a lightweight draft model and a larger base model, aiming to minimize latency while preserving accuracy (see Algorithm 1). We refer to this framework as *HoVer*, short for *Holistic Verification*, since it leverages semantic-level information to judge correctness holistically across text-spans rather than token by token. The framework consists of three stages: (1) draft generation, where the draft model produces a complete candidate output; (2) error detection, where the base model evaluates the draft and determines the longest correct prefix; and (3) output rewriting, where the base model resumes generation from that prefix if errors are detected. Fig. 2 illustrates this process, showing how the draft model may produce errors and how the base model verifies and corrects them.

**Stage 1: draft generation.** Given the prompt, the draft model first produces a complete candidate answer (line 3 of Alg. 1). In our experiments, draft models range from 1.7B to 8B parameters and can run on a single GPU, or as a co-located sidecar alongside the base-model serving stack with modest

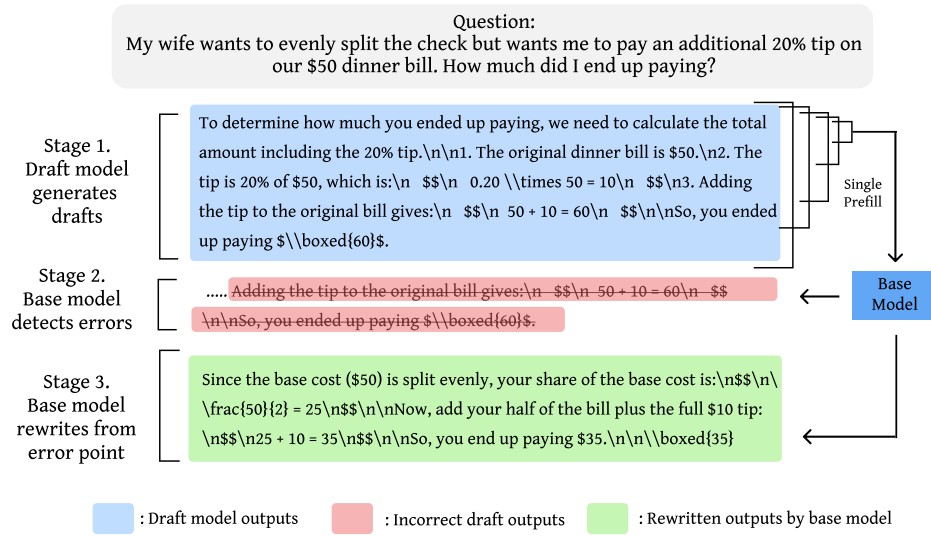

Figure 2: This is a concrete example of our framework. The draft model makes mistakes starting from the fourth line (the strikethrough red-block text) and gives the wrong answer 60. The revised model regenerates from the error point, giving us the right answer 35.

additional capacity. On our cluster, these draft models decode at roughly $\sim 10\times$ the throughput of the large MoE base model (i.e., Qwen3 235B MoE). Hence, if the base model accepts even $\approx 10\%$ of the draft tokens, end-to-end latency matches the baseline; higher acceptance yields net speedup.

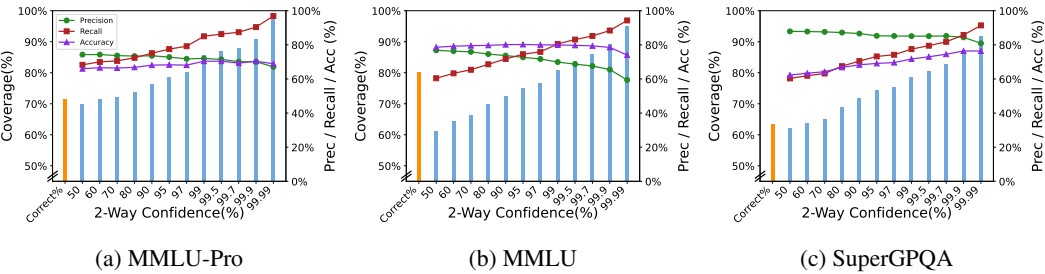

(a) MMLU-Pro  (b) MMLU  (c) SuperGPQA

Figure 3: Confidence threshold $\tau$ and error-identification coverage with the base model `Qwen3 235B`. The 2-way confidence is defined as $P(\text{Correct})/(P(\text{Correct}) + P(\text{Incorrect}))$. The base model accepts a draft only if this confidence exceeds $\tau$. Coverage denotes the fraction of the draft model's errors that the base model successfully detects. Answers marked `Incorrect` are treated as positives, and precision, recall and accuracy are evaluated. The orange bar shows, among the draft model's mistakes, how often the base model produces the correct answer.

**Stage 2a: Error Detection [Full Answer Verification]** The base model makes two decisions: (1) whether the draft answer can be accepted (line 4 in Alg. 1), and (2) if not, which prefix remains valid (line 9 in Alg. 1). The first decision is performed in a *prefill-only* manner (Du et al., 2025): the revise model prefills the entire draft once and emits a single classification token, `Correct` or `Incorrect`. Because no step-wise decoding occurs, there is no need to maintain a key–value (KV) cache; hence, verification is compute-bound and incurs minimal additional memory.

We observe in Fig. 3 that using the `Correct`/`Incorrect` token alone to judge the draft model's response correctness (which corresponds to 2-way confidence $= 50\%$ in Fig. 3) is insufficient, as only $\sim 60 - 70\%$ of errors are identified. By setting the confidence threshold $\tau$ higher, we observe that the base model's judgment ability can go beyond its answering ability (orange bars in Fig. 3), identifying errors even in cases where the base model itself cannot answer correctly, maximizing the

chances of fixing the errors in the revision phase later. After testing on a small validation set, we set $\tau = 99.7\%$ to strike a balance between accuracy and efficiency. [Peter: New]

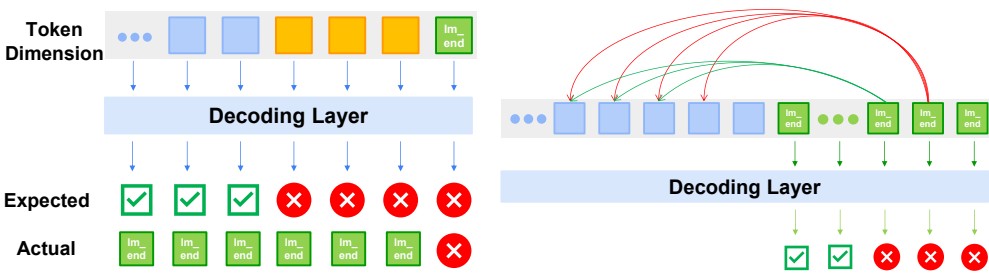

(a) Segmenting at the token dimension outputs template tokens instead of Correct/Incorrect tokens

(b) Appending template tokens at the end to attend to different prefixes solves the problem.

Figure 4: Using Token Level Segmentation to Predict Prefix Correctness

---

**Algorithm 1:** *HoVer*: Stage 1 (Draft) $\rightarrow$ Stage 2 (*Holistic* Verify) $\rightarrow$ Stage 3 (Revise)

1 **Parameters:** confidence threshold $\tau$; chunk step $\Delta$.
2 **foreach** *question q* **do**
    // ▷ Stage 1:  Draft Generation
3     $y_s \leftarrow$ SMALLDRAFT$(q)$
    // ▷ Stage 2a:  Error Detection [Full Answer Verification]
4     $(v, \text{conf}) \leftarrow$ JUDGEFULL$(q, y_s)$
5     mode $\leftarrow$ undefined;  prefix $\leftarrow \varnothing$
6     **if** $v = $ *Correct* **and** $\text{conf} \geq \tau$ **then**
7       | mode $\leftarrow$ accept-small
    // ▷ Stage 2b:  Error Detection [Prefix Verification]
8     **if** *mode* $=$ undefined **then**
9       $\chi \leftarrow$ VERIFYPARTIAL$(q, y_s; \Delta)$
10       $j^\star \leftarrow \chi.$last_correct_chunk
11       **if** $j^\star \neq \varnothing$ **then**
        // ▷ Form a safe prefix ending at the identified boundary
12         prefix $\leftarrow$ PREFIXUNTIL$(y_s, j^\star)$
13         mode $\leftarrow$ continue
14       **else**
        // ▷ No reliable prefix identified; fall back to restart
15         mode $\leftarrow$ restart
    // ▷ Stage 3:  Output Rewriting
16     **if** *mode* $=$ accept-small **then**
17       **accept** $y_s$     // ▷ Case 1:  full answer correct (confidence $\geq \tau$)
18     **else if** *mode* $=$ continue **then**
19       $y_{\text{cont}} \leftarrow$ BIGCONTINUE$(q, \text{prefix})$     // ▷ Case 2:  continue from the prefix
20       $y_{\text{final}} \leftarrow$ prefix $\parallel y_{\text{cont}}$
21       **accept** $y_{\text{final}}$
22     **else if** *mode* $=$ restart **then**
23       $y_b \leftarrow$ BIGFROMSCRATCH$(q)$     // ▷ Case 3:  regenerate from scratch
24       **accept** $y_b$

---

**Stage 2b: Error Detection [Prefix Verification]** When the draft answer is judged incorrect, the base model estimates the longest prefix that can be safely reused. A naive approach would be to run prefill-only verification on many truncated prefixes one by one. This approach's cost scales linearly with the number of verified prefixes and is slow for long answers due to its sequential nature.

We note that every partial answer is itself a prefix of the full draft sequence. Leveraging the causal nature of attention in decoder-only LLMs, we can evaluate the correctness of multiple partial answers in a single prefill. Specifically, we prefill the complete answer and segment along the token

dimension before the final decoding layer, as shown in Fig. 4a. Instead of decoding one position, we decode $n$ tokens simultaneously, where $n$ is the number of prefixes to verify.

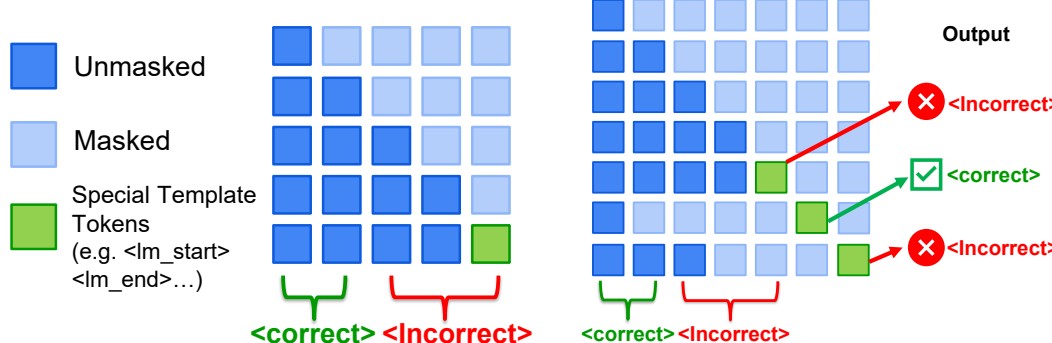

(a) Original causal attention mask. Special template tokens only attend to the entire prefix

(b) Custom attention mask enables efficient prefix correctness identification with one prefill pass

Figure 5: Comparison between the original and our custom attention mask

However, naive truncation breaks the input format expected by the model. In decoder-only LLMs, classification tokens such as `Correct`/`Incorrect` are only produced when the input ends with the proper chat-template suffixes (e.g., `lm_start`, `assistant`, `lm_end`). If we simply cut the draft at the token dimension and feed these truncated prefixes directly into the model, the required suffix is missing. As a result, the model defaults to emitting the missing template tokens such as `lm_end` rather than the intended correctness labels (Fig. 4a).

To fix this, we augment every truncated prefix with a duplicated chat-template *suffix*, forming complete and valid sequences (Fig. 4b). A custom attention mask then ensures that each suffix only attends to its paired prefix (Fig. 5b). With this construction, the base model can output $n$ independent `Correct`/`Incorrect` judgments in a single forward pass. This procedure corresponds to `VerifyPartial` in line 9 of Alg. 1. Once the last correct position is identified, generation continues from there using the larger model.

**Stage 3: Output Rewriting**   Given the verification outcomes, the *HoVer* framework chooses one among the three following paths: **(1) Accept the draft** if the full-answer judgment is `Correct` with confidence $\geq \tau$ (line 6–7, 16–17 in Alg. 1). **(2) Continue from a prefix** if a longest reliable prefix is identified; the revise model continues generation from that prefix, avoiding redundant recomputation. (Line 12–13, 18–21 in Alg. 1) **(3) Regenerate from scratch** with the base model if no prefix is deemed safe, or if all prefix is judged as correct, and we cannot identify a clear beginning of error(Line 14–15, 22–24 in Alg. 1).

### 3.2   COMPATIBILITY WITH TOKEN-LEVEL SPECULATIVE DECODING

Our semantic-level *HoVer* framework is orthogonal to previous token-level speculative methods and can be combined with them for additional speedups. In particular, token-level speculative decoding may be applied during draft generation (stage 1) and during the base model's output rewriting (stage 3), compounding gains without changing our verification mechanism.

## 4   EXPERIMENTS

### 4.1   SETUP

We conduct the experiments with state-of-the-art open-source chat models. For draft model, we use `Qwen3 1.7B` and `Qwen3 8B`. For base model, we use `Qwen3 235B MoE Insturct` (Yang et al., 2025a). To demonstrate that our method is robust across model families, we further add a case study where we use `Llama3.3 70B` (Grattafiori et al., 2024) as the base model, and `Qwen4 4B` as the draft model. We did not choose `Llama3.1 8B` or smaller models from the Llama family as the

Table 1: Baseline vs. *HoVer* Accuracy (%), Speed (tokens/s) and Speedups (***Qwen 3***)

| Size | Method | MMLU Redux | MMLU Pro | Super GPQA | GSM8K | MATH | GPQA Diamond | HumanEval | MBPP |
|------|--------|------------|----------|------------|-------|------|--------------|-----------|------|
| 235B | Baseline | 87.80 | 69.76 | 56.75 | 92.12 | 80.08 | 52.02 | 85.40 | 93.70 |
| +8B | HoVer | 84.47 | 67.62 | 52.00 | 93.18 | 79.86 | 50.51 | 90.20 | 90.50 |
| +1.7B | HoVer | 85.47 | 66.19 | 53.25 | 90.75 | 77.62 | 51.01 | 79.90 | 87.30 |
| 8B | Draft Only | 78.47 | 60.71 | 33.25 | 92.04 | 77.52 | 43.94 | 76.20 | 71.40 |
| 1.7B | Draft Only | 62.17 | 40.95 | 25.75 | 77.33 | 63.74 | 31.31 | 53.70 | 33.30 |
| *Speed (tokens/s) and Speedup* | | | | | | | | | |
| 235B | Baseline | 6.71 | 6.88 | 6.87 | 6.49 | 6.85 | 6.94 | 6.34 | 5.89 |
| +8B | HoVer | 15.82 | 8.23 | 9.23 | 20.62 | 11.08 | 8.24 | 7.45 | 6.72 |
| | *Speedup* | 2.35× | 1.19× | 1.34× | 3.18× | 1.61× | 1.19× | 1.17× | 1.14× |
| +1.7B | HoVer | 10.52 | 9.20 | 9.02 | 17.98 | 10.25 | 8.42 | 6.32 | 5.57 |
| | *Speedup* | 1.56× | 1.34× | 1.31× | 2.77× | 1.49× | 1.21× | 1.00× | 0.94× |

*Note:* "235B Baseline" is directly using the Qwen3-235B MoE base model to generate responses. "+8B / +1.7B HoVer" denotes *HoVer* with Qwen3-8B / Qwen3-1.7B as the draft model and Qwen3-235B as the base model. "8B / 1.7B" are draft-only baselines (no base model verification and output rewriting).

Table 2: Baseline vs. *HoVer* Accuracy (%), Speed (tokens/s) and Speedups (***Llama 3.3***)

| Size | Method | MMLU Redux | MMLU Pro | Super GPQA | GSM8K | MATH | GPQA Diamond | HumanEval | MBPP |
|------|--------|------------|----------|------------|-------|------|--------------|-----------|------|
| 70B | Baseline | 78.00 | 67.14 | 36.2 | 93.86 | 67.72 | 50.51 | 84.10 | 87.60 |
| +4B | HoVer | 76.33 | 63.86 | 35.75 | 91.96 | 72.67 | 48.48 | 73.20 | 87.30 |
| 4B | Draft Only | 73.00 | 56.73 | 32.50 | 89.08 | 72.94 | 40.40 | 65.20 | 73.80 |
| *Speed (tokens/s) and Speedup* | | | | | | | | | |
| 70B | Baseline | 18.46 | 18.39 | 18.36 | 18.52 | 18.48 | 18.38 | 18.14 | 18.12 |
| +4B | HoVer | 34.60 | 27.29 | 19.68 | 40.55 | 36.29 | 19.71 | 25.15 | 19.20 |
| | *Speedup* | 1.87× | 1.48× | 1.07× | 2.18× | 1.96× | 1.07× | 1.38× | 1.05× |

*Note:* "70B Baseline" is directly using the Llama3.3-70B-Instruct base model to generate responses. "+4B HoVer" denotes *HoVer* with Qwen3-4B as the draft model and Llama3.3-70B as the base model. "4B" is draft-only baselines (no base model verification and output rewriting).

base model, because our method's performance relies on a capable small model. A small model that makes too mistakes will only slow down the generation speed. We implement the *HoVer* framework and an efficient partial verification algorithm on top of Huggingface transformers library (Wolf et al., 2020).

We compare with SpecReason (Pan et al., 2025) by adapting it for general text generation tasks. Our results show that compared to prior semantic-aware speculative generation work, the holistic nature of our methods is a key enabler of better generation quanlity and faster generation speed. We do not directly compare with (Bachmann et al., 2025) as their training requires undisclosed datasets, and their method can be viewed as a high-performance token-level speculative decoding method, that can be combined with ours.

We do experiments on three types of tasks: **General Tasks** (MMLU-Redux (Gema et al., 2025), MMLU-Pro (Wang et al., 2024), SuperGPQA (Team et al., 2025)), which are multi-choice questions covering a wide range of disciplines, **Math & STEM tasks** (MATH (Hendrycks et al., 2021), GSM8K (Cobbe et al., 2021), GPQA (Rein et al., 2024)), and **Coding Tasks** (HumanEval (Chen et al., 2021), MBPP (Austin et al., 2021)). During our evaluation, we directly use the instruct mode of the LLM, with no CoT or reasoning process enabled. For multi-choice datasets such as MMLU and GPQA, we ask the model to generate explanation and justification before giving the final answer, instead of outputting a single letter of choice. We run the experiments on 4 `NVIDIA H200 GPUs`, interconnected via `NVLink`.

We evaluate our method using the following metrics: **(1) Speedup Ratio**, i.e., the latency reduction relative to having the base model directly answer the question. **(2) Accuracy.** The accuracy of *HoVer* on downstream tasks and benchmarks.

### 4.2 ACCURACY AND SPEEDUP

Table 2 compares the accuracy of *HoVer* against directly using the base or draft models. It shows that our *HoVer* framework has comparable accuracy with much larger MoE models, and improves

Table 3: SpecReason vs. *HoVer* Accuracy (%) and Speed (tokens/s)

| Metric | SuperGPQA | GPQA Diamond | MATH | GSM8K |
|---|---|---|---|---|
| SpecReason Accuracy | 42.59 | 41.41 | 72.67 | 94.00 |
| *HoVer* Accuracy | 52.00 | 50.51 | 79.86 | 93.18 |
| SpecReason Speed (tok/s) | 8.37 | 7.34 | 8.83 | 8.87 |
| *HoVer* Speed (tok/s) | 9.23 | 8.24 | 11.08 | 20.62 |

significantly on the draft model's accuracy, showing that the improvement is inherent in the design itself, rather than coming from the power of smaller models. Further, it demonstrates that our method is robust regardless of the choice of draft models. We can even use a draft model whose parameter size is as small as 1.7B, and still retain comparable accuracy on all tested datasets. This is because our base model is able to robustly identify the beginning of errors and make amendments, regardless of the draft model's accuracy.

We further analyze the speedup ratio on different tasks. As shown in Table 2, the results vary depending on the choice of the draft models and the difficulty level of the datasets. Regardless of the difficulty level, our methods accelerate the generation by $1.19\times$ minimum, and up to $3.18\times$. Overall, we observe the following trends:

- Larger draft models have better speedup. This is because the larger draft model's output has a larger chance of being partially or totally accepted by the bigger model.
- Simpler tasks offer more significant improvement. Simple datasets like GSM8K yield a speedup of more than 200%, while challenging datasets like SuperGPQA or GPQA-Diamond yield a speedup of $\sim 19\% - 30\%$.
- The acceleration is more salient with longer generations. For dataset with shorter generation length such as HumanEval and MBPP, where usually only tens of tokens are generated, the benefit of small model generation and holistic verification is dominated by the overhead.

We test thoroughly on many datasets, ranging from easier datasets such as GSM8K to challenging ones like MMLU-Pro and SuperGPQA. Our method consistently yields comparable accuracy to larger base models and offers speedup. This shows that our method is robust in math and general tasks, regardless of the inherent difficulty level of the questions themselves.

### 4.3 COMPARISON WITH OTHER SEMANTIC LEVEL GENERATION METHODS

We compare with existing semantic-aware speculative generation method such as SpecReason (Pan et al., 2025) in Table 3, by adapting its code for general text generation, instead of chain-of-thoughts generation. SpecReason generates some text, and judge at fixed points whether to accept the last generated text segment from the small model, or rewrite that segment with the big model, based on a 1 to 10 score given by the big model.

We find that our method excels SpecReason in both the generation quality and speculation speed. We hypothesize the reasons to be: (1) SpecReason lacks holistic verification and correction, while chunk by chunk partial correction along the generation trajectory is insufficient to fix all errors, causing a degradation of generation quality; (2) the sequential verification process of SpecReason adds much overhead to the generation, lowering generation speed; (3) the score-based judgment is less robust than our confidence-based method, leading to lower generation speed even when the questions are simple (e.g. GSM8K). In comparison, HoVer adapts to the inherent difficulty level of the question itself, yielding more speedup when answering simpler questions.

### 4.4 ROBUSTNESS OF TWO-WAY CONFIDENCE ERROR IDENTIFICATION

We evaluate the robustness of using 2-way confidence as a metric for correctness classification, by assessing the error coverage, precision, accuracy and recall under different confidence threshold setting as in Figure 3. Answers with confidence lower than the threshold are treated as `Incorrect` (positives). With higher confidence threshold, HoVer consistently yields higher recall, while maintaining comparable precision and accuracy. This shows that with 2-way confidence, HoVer can robustly identify errors successfully, without causing too many false positives.

## 4.5 ACCELERATION BREAKDOWN

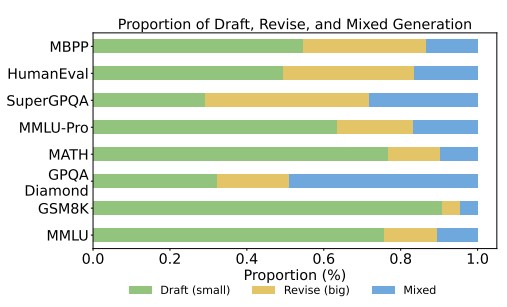

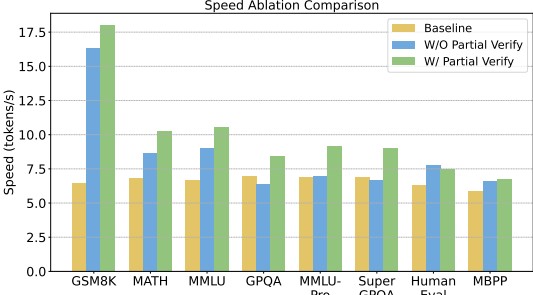

(a) Final response source breakdown: *Draft* is the draft model's response being accepted; *Revise* is using base model to regenerate; *Mixed* is base model continue generating from partial draft

(b) Speed with Qwen3-235B Base Model only, *HoVer* without partial verification, and *HoVer* with partial verification

Fig. 6a breaks down the sources of final responses, showing the proportion of answers that come from drafts directly accepted, full regeneration by the base model, or continuation from a partial draft. The distribution varies substantially across datasets. On easier benchmarks such as GSM8K, a large fraction of responses are accepted directly from the draft model, yielding significant speedups. On more challenging datasets such as GPQA-Diamond, fewer drafts can be accepted outright; however, *HoVer* effectively reuses partial drafts by continuing from the last correct prefix, enabling faster regeneration compared to restarting from scratch.

We observe that among all the draft model's answers rejected by the base model, around half of those responses are reused with our efficient partial verification and output rewriting. This shows the importance of introducing the efficient partial verification technique.

## 4.6 ABLATION EXPERIMENT

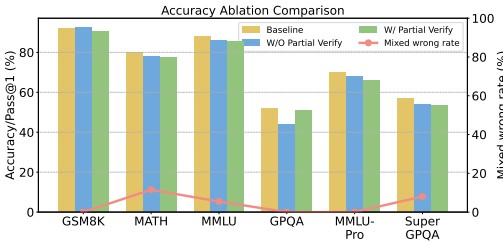

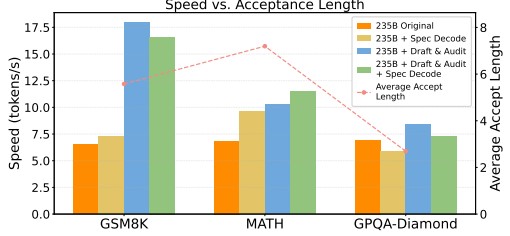

Figure 7: Accuracy with Qwen3-235B, *HoVer* without partial verification, and *HoVer* with partial verification

Figure 8: *HoVer* used in conjunction with token-level speculative decoding, and the average accept length of speculative decoding across datasets

We conduct an ablation study to isolate the contributions of the draft model and partial verification.

**Speed.** The yellow bars in Fig. 6b show the baseline generation speed of the Qwen3 235B MoE model. The green bar corresponds to our method, where Qwen3 235B is used as the base model and Qwen3 1.7B as the draft model. The blue bar removes partial verification: whenever the revise model rejects the draft response, generation restarts from scratch with the larger base model, rather than resuming from the last correct prefix, which is in essence a variant of model cascade. From Fig. 6b, we observe that partial verification is crucial for overall speedup; without it, the acceleration is much smaller.

**Accuracy.** Fig. 7 reports accuracy for the same three cases as in Fig. 6b. Partial verification can correctly identify the position of errors, ensuring that performance does not degrade compared to the baseline. The Mixed Wrong Rate in Fig. 7 quantifies how often the revision stage introduces an error in cases where the base model, under direct prompting, would have produced the correct

answer. It remains low across datasets, showing that partial verification and revision do not harm the base model's inherent accuracy.

**Combination with token-level speculative decoding.** We also demonstrate how token-level speculative decoding can be combined with *HoVer* for further speedup. We adopt Hugging Face's implementation of assistant speculative decoding (Timor et al., 2025). As shown in Fig. 8, the benefit depends strongly on the dataset. For MATH, token-level speculative decoding achieves a large acceptance length, and thus combining *HoVer* with token-level speculation yields additional acceleration beyond both the baseline 235B inference and *HoVer*. For GSM8K and GPQA-Diamond, however, acceptance lengths of token-level decoding are too small to amortize the overhead, resulting in slower generation than using *HoVer* alone.

### 4.7 CASE STUDIES

We manually inspect 200 cases from the experiment results where the draft model's response is partially correct, and the larger model is able to identify the beginning of the errors and continue from the last correct position. We show one such example here.

In this example, black text shows the shared prefix; red marks the first incorrect span found by the base model in a single pass; green text is the regenerated suffix. *HoVer* then outputs black + green text as the final answer, demonstrating that the base model reliably locates the last correct position and efficiently resumes generation, maximizing token reuse.

---

**Question:** Kenny is selling his *Pokemon* cards to buy a ticket to an amusement park, which costs $100. He has a collection of cards and plans to sell them for $1.5 each. He keeps 1/3 of them and gets to go to the amusement park with $50 in spending cash. How many cards did he start with?

---

Let's break down the problem step by step. Kenny wants to buy a ticket to an amusement park that costs $100. He has a collection of *Pokemon* cards, and he plans to sell them for $1.50 each. He keeps 1/3 of the cards, meaning he sells 2/3 of them. After selling the cards, he has $50 in spending cash. We are to find out how many cards Kenny started with. Step 1: Let the total number of cards be $x$. He sells $\frac{2}{3}x$ cards. Each card is sold for $1.50, so the total money he gets from selling the cards is: *Money from selling cards* $= 1.5 \times \frac{2}{3}x = 1.0x$. So, he earns $x from selling the cards. He has $50 in spending cash, and he needs $100 for the ticket. That means he needs an additional $50. So, the money he earned from selling the cards must be $50: $x = 50$ — Final Answer: 50. Kenny started with 50 cards. the cards. This money is used to buy a $100 ticket and leaves him with $50 in spending cash. So the total money he earns from selling the cards is: $100 + $50 = $150. We already found that the money earned is $x$ dollars (since $1.5 \times \frac{2}{3}x = x$). So: $x = 150$. Therefore, Kenny started with 150 *Pokemon* cards.

---

## 5 CONCLUSION

We presented *HoVer*, a semantic-level speculative generation framework based on *holistic verification*. *HoVer* reduces LLM latency without retraining or altering the base model. The core ideas include using larger models to perform semantic-level identification of wrong answers, and a *prefill-only*, *parallel* verification of multiple prefixes in a single forward pass using a custom attention mask, keeping verification compute-bound with negligible KV-cache overhead. Across model families and datasets, we observe $\sim1.2\times$–$3.1\times$ speedups while maintaining base-model accuracy; ablations show that partial verification is a primary driver by enabling continuation from the last safe prefix. The method is orthogonal to token-level speculation and can be used in conjunction. Overall, *HoVer* reframes speculative generation around semantics through holistic verification and offers a practical path to faster and reliable LLM serving.

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

## A  THE USE OF LARGE LANGUAGE MODELS (LLMs)

Large language models (LLMs) were used solely to polish the writing, improve fluency, and correct grammatical errors. No part of the research ideation, methodology, experiments, analysis, or substantive writing relied on LLMs. The authors take full responsibility for the content of this paper.

## B  PROMPT FOR STAGE 2 OF ALGORITHM 1

In Stage 2a (*Error Detection [Full Answer Verification]*), the base model holistically evaluates whether the draft answer is correct (see Algorithm 1, line 4). This verification is conducted in a *prefill-only* manner: the entire draft is passed once through the model, which then emits a classification token (`Correct` or `Incorrect`). This design keeps verification compute-bound, avoids KV-cache growth, and allows correctness to be judged at the semantic level. The following prompt specifies the verification instruction used by the model in this stage.

---

**Stage 2a Verification Prompt**

**SYSTEM PROMPT**
You are a cold, conservative, strict and rigorous verifier of COMPLETE solutions. Given a question and a proposed answer, decide if the answer is correct. There are two modes—pick exactly one: • Math: math problems/derivations/numeric computations. • Science-QA: science multiple-choice (A–J) style questions. Respond with only one word: Correct or Incorrect. You are strict and cautious. If there is doubt, answer Incorrect.

**RUBRIC PROMPT**
=== MODE PICKER (deterministic) === Use Science-QA iff the question or answer clearly involves options A–J (e.g., an explicit choice letter). Otherwise use Math. Do NOT switch modes mid-judgment.
=== GLOBAL STRICTNESS (applies to both) === Judge ONLY the provided content. If the final conclusion is missing, unclear, or contradictory → Incorrect. If multiple conflicting final conclusions exist → Incorrect. If any decisive error exists → Incorrect. When in doubt → Incorrect.
=== MATH MODE RULES === Check ALL: (1) arithmetic/algebra identities and steps are valid; (2) domain/units/constraints respected (e.g., probabilities in [0,1], counts are non-negative integers); (3) internal consistency; (4) completeness: an explicit final answer that matches what is asked. Numeric answers must be exactly correct unless a tolerance is explicitly justified; symbolic answers must be algebraically equivalent. Any failure → Incorrect.
=== SCIENCE-QA MODE RULES === You will see a scientific question and a proposed answer/explanation that may include a final choice letter (A–J). Use ONLY the provided content and commonly accepted scientific facts; do not invent options you cannot infer. Guidelines: 1) If the proposed answer clearly concludes with a single choice (A–J or equivalent like \boxed{C}), judge that conclusion. 2) If multiple conflicting final choices are present, or no conclusion is given, answer Incorrect. 3) If the reasoning is obviously wrong or contradicts established science relevant to the question, answer Incorrect. 4) If the reasoning is sound and the conclusion plausibly follows, answer Correct. 5) Be cautious but not adversarial; when evidence strongly supports the given conclusion, answer Correct.
=== OUTPUT FORMAT (strict) === Reply with EXACTLY one word on the first line: Correct or Incorrect

---

In Stage 2b (*Error Detection [Prefix Verification]*), the base model judges whether each *prefix* of the draft answer remains valid, allowing *HoVer* to reuse the longest correct prefix rather than restarting from scratch (see Algorithm 1, line 9). This is implemented via a single prefill-only forward pass that evaluates multiple prefixes in parallel using a custom attention mask (Fig. 5b). The following prompt specifies the instructions used to guide the verifier when checking partial prefixes.

---

**Stage 2b Partial Verification Prompt**

**JUDGE_SYSTEM**
You are a strict but fair verifier of PARTIAL solutions. You see a question and an in-progress answer prefix. Your job is to judge ONLY the shown steps/claims so far, not the unseen remainder. There are two modes: • Math-Partial: mathematical derivations/solutions. • QA-Partial: science multiple-choice (A–J) style questions with domain facts. Pick the mode that fits the question and judge accordingly.

**RUBRIC PROMPT**
=== MODE SELECTION === Choose ONE mode based on the question: • Use Math-Partial if the task is a math problem/derivation or numeric proof-style reasoning. • Use QA-Partial if the task is a scientific multiple-choice question (A–J style) or scientific reasoning. Do NOT switch modes mid-judgment.

=== Math-Partial STANDARD === **Scope:** Evaluate ONLY the shown work-in-progress. Do NOT assume missing steps or future fixes. *Lenient allowances (do NOT penalize):* harmless notation quirks, skipped trivial algebra, rounding that does not affect claims. *Strict errors (mark Incorrect):* arithmetic/algebra mistakes, invalid operations, domain violations, logical contradictions. *Tie-breaker:* If no concrete error is visible in the SHOWN steps, output Correct.

=== QA-Partial STANDARD === **Scope:** Evaluate ONLY the shown reasoning relative to the stem. *Lenient allowances (do NOT penalize):* lack of final choice, exploratory reasoning, benign phrasing. *Strict errors (mark Incorrect):* factual falsehoods, quantitative mistakes, contradictions with the stem, invalid inferences, or explicit commitment to a ruled-out option. *Neutral/incomplete:* If no falsifiable claim is made yet, output Correct. *Tie-breaker:* Absent a definite error in the SHOWN text, output Correct.

=== OUTPUT FORMAT (STRICT) === Reply with EXACTLY one word on the first line: Correct or Incorrect No other text is allowed.

---

## C EXAMPLES OF CONTINUAL GENERATION FROM THE LAST CORRECT PREFIX

In this section, we provide illustrative case studies showing how *HoVer* leverages prefix verification to recover from draft model errors. Specifically, when the draft model deviates from the correct solution, the base model is able to identify the earliest point of error and resume generation from the last valid prefix, rather than restarting from scratch. These examples highlight the effectiveness of continual generation: by reusing correct portions of the draft, *HoVer* minimizes redundant computation while still ensuring accurate final answers. We follow the same convention as in section 4.7, where black text is the reused prefix, red text is the rejected portion of the draft, and the green text is the new suffix regenerated by the base model.

---

**Question:** The largest exporter of agricultural goods in the world is :A. the United States.B. Mexico. C. Chile. D. China?

---

To determine the largest exporter of agricultural goods in the world, we need to consider the global agricultural trade landscape. 1. United States (A): The U.S. is a major agricultural producer and exporter, particularly in crops like corn, soybeans, and wheat. However, it is not the largest exporter globally in terms of total value. 2. Mexico (B): Mexico is a major exporter of agricultural goods, especially in fruits, vegetables, and dairy products. It is one of the largest agricultural exporters in the world, but it is not the largest. 3. Chile (C): Chile is a major exporter of agricultural goods, particularly in wine, beef, and other high-value products. It is one of the top agricultural exporters in the world. 4. China (D): China is the largest exporter of agricultural goods in the world, with significant exports of rice, wheat, and other staple crops. It is the largest in terms of total value and volume. Final Answer: D a significant exporter of fruits, wine, and seafood, particularly to North America and Asia. 4. China (D): China is the world's largest agricultural producer, but it is not a

top net exporter. In fact, the United States consistently ranks as the largest exporter of agricultural goods by total value, according to data from the USDA and the World Trade Organization. While other countries may lead in specific commodities, the U.S. leads in overall agricultural export value. Final Answer: A

---

**Question:** A custodian has to clean a school with 80 classrooms. They have 5 days to get it done. It takes them 15 minutes per classroom. If they work an 8 hour day, what percentage of their day, on average, is spent cleaning classrooms?

---

To find the percentage of the custodian's day spent cleaning classrooms, we need to follow these steps: Step 1: Total time spent cleaning classrooms - There are 80 classrooms. - It takes 15 minutes per classroom. - Total time spent cleaning = $80 \times 15 = 1200$ minutes. Step 2: Convert total time to hours - 1200 minutes $= \frac{1200}{60} = 20$ hours. Step 3: Total time available in a day - The custodian works 8 hours per day. Step 4: Calculate the percentage of the day spent cleaning - Percentage $= \left(\frac{20}{8}\right) \times 100 = 250\%$. Final Answer: 250% Step 3: Total available work hours - The custodian works 8 hours per day for 5 days. - Total available work time = $8 \times 5 = 40$ hours. Step 4: Calculate percentage of time spent cleaning - Time spent cleaning = 20 hours. - Total work time = 40 hours. - Percentage of time spent cleaning $= \left(\frac{20}{40}\right) \times 100 = 50\%$. The custodian spends 50% of their time, on average, cleaning classrooms. Final Answer: $50\%$

---

**Question:** Hannah's city is having a big display of fireworks for the 4th of July. They are going to set off 15 boxes of 20 fireworks each. Hannah's house is at the right angle to see 40% of the city's fireworks. Hannah will also set off 3 boxes of 5 fireworks each in her backyard. How many fireworks will Hannah see in total?

---

To find out how many fireworks Hannah will see in total, we need to break the problem into parts: Step 1: Total number of fireworks set off in the city - There are 15 boxes, and each box contains 20 fireworks: $15 \times 20 = 300$ fireworks. Step 2: Number of fireworks Hannah sees from the city display - Hannah can see 40% of the city's fireworks: $0.40 \times 300 = 120$ fireworks. Step 3: Number of fireworks Hannah sets off in her backyard - Hannah sets off 3 boxes, and each box contains 5 fireworks: $3 \times 5 = 15$ fireworks. Step 4: Total number of fireworks Hannah sees - Hannah sees the 120 fireworks from the city display and also sets off 15 fireworks in her backyard. However, the question asks how many fireworks Hannah will see in total, which refers only to the ones she sees, not the ones she sets off. So, the total number of fireworks Hannah sees is: 120 (from the city) $+ 0$ (from her own fireworks, since she sets them off, not sees them) $= 120$. Final Answer: 120 fireworks Hannah sees include both the city display fireworks visible from her house and the fireworks she sets off in her backyard, since she would presumably watch the ones she sets off. Therefore, we add the two quantities: 120 (from city) $+ 15$ (from backyard) $= 135$ fireworks. Hannah sees a total of 135 fireworks. $\boxed{135}$

---

**Question:** There are 4 snails in one aquarium and 32 snails in another aquarium. The difference between the number of snails in the two aquariums is twice the amount of fish in both aquariums. If both aquariums have the same number of fish in them, how many fish are there in each aquariumThere are 4 snails in one aquarium and 32 snails in another aquarium. The difference between the number of snails in the two aquariums is twice the amount of fish in both aquariums. If both aquariums have the same number of fish in them, how many fish are there in each aquarium?

---

We are given the following information: - There are 4 snails in one aquarium. - There are 32 snails in another aquarium. - The difference between the number of snails in the two aquariums is twice the amount of fish in both aquariums. - Both aquariums have the same number of fish. Step 1: Find the difference in the number of snails $32 - 4 = 28$. So, the difference in the number of snails is 28. Step 2: Use the given relationship We are told that this difference (28) is twice the amount of fish in

both aquariums. Let the number of fish in each aquarium be $x$. Then, the total number of fish in both aquariums is: $x + x = 2x$. We are told: $2x = 28$. Step 3: Solve for $x$ $x = \frac{28}{2} = 14$. Final Answer: 14 each aquarium be $f$. Since both aquariums have the same number of fish, the total number of fish in both aquariums is: $f + f = 2f$. According to the problem, the difference in snails (28) is twice the total number of fish in both aquariums: $28 = 2 \times (2f)$. Simplify the right side: $28 = 4f$. Now solve for $f$: $f = \frac{28}{4} = 7$. Final Answer: 7. There are 7 fish in each aquarium.

---

**Question:** What is the traditional 20th wedding anniversary gift? Options: A. silver B. emerald C. pearl D. diamond E. leather F. sapphire G. ruby H. paper I. china J. gold

---

To determine the traditional 20th wedding anniversary gift, we need to refer to the traditional gift associations for wedding anniversaries. Here is the traditional gift list for wedding anniversaries: - 1st: Paper - 2nd: Cotton - 3rd: Leather - 4th: Fruit - 5th: Wood - 6th: Iron - 7th: Wool - 8th: Cotton - 9th: Cotton - 10th: Tin - 11 th: Copper - 12th: Fabric - 13th: Fabric - 14th: Crystal - 15th: Crystal - 16th: Lapis Lazuli - 17th: Lapis Lazuli - 18th: China - 19th: China - 20th: Silver. So, the traditional 20th wedding anniversary gift is silver. Final Answer: $\boxed{A}$ 11th: Steel - 12th: Silk - 13th: Lace - 14th: Ivory - 15th: Crystal - 20th: China. The traditional 20th wedding anniversary gift is china. $\boxed{I}$

---

**Question:** What is the largest order of an element in the group of permutations of 5 objects? Options: A. 12 B. 6 C. 5 D. 15

---

To determine the largest order of an element in the group of permutations of 5 objects, we need to understand the structure of the symmetric group $S_5$. Step 1: Understand the group The symmetric group $S_n$ consists of all permutations of $n$ objects. The order of a permutation is the least common multiple (LCM) of the lengths of its disjoint cycles. Step 2: Find the largest possible cycle length In $S_5$, the maximum possible cycle length is 5 (a single 5-cycle). The order of a 5-cycle is 5, since the LCM of 5 is 5. Step 3: Consider other cycle types Other cycle types in $S_5$ include: - 4-cycles (order 4) - 3-cycles (order 3) - 2-cycles (order 2) - 1-cycles (order 1). The largest order among these is 5 (from a 5-cycle). Step 4: Confirm the answer The largest order of an element in $S_5$ is 5. Final Answer: $\boxed{C}$ Products of disjoint cycles, such as a 3-cycle and a 2-cycle (disjoint), which gives a permutation of type (3,2). We must consider all integer partitions of 5 and compute the LCM of the cycle lengths. Partitions of 5 and corresponding LCMs: 1. (5) → LCM(5) = 5 2. (4,1) → LCM(4,1) = 4 3. (3,2) → LCM(3,2) = 6 4. (3,1,1) → LCM(3,1,1) = 3 5. (2,2,1) → LCM(2,2,1) = 2 6. (2,1,1,1) → LCM(2,1,1,1) = 2 7. (1,1,1,1,1) → identity, order 1. The maximum among these is LCM(3,2) = 6. Is there a partition that gives a higher LCM? Try (3,2): 3+2=5 → valid, LCM=6. (5): LCM=5. (4,1): 4. (2,3): same as (3,2). No partition gives LCM higher than 6. Wait: is there a way to get LCM = 12? That would require cycles of length 3 and 4, but 3+4=7 ¿ 5 — impossible. LCM = 15? Needs 3 and 5 → 3+5=8 ¿ 5 — impossible. So maximum is 6. But wait — is there a 5-cycle? Order 5. Or a 3-cycle and a disjoint 2-cycle: yes, in $S_5$, we can have (1 2 3)(4 5). This has order LCM(3,2) = 6. No other combination gives higher. Thus, the largest order is 6. $\boxed{B}$

