# OpenReview forum: "HoVer: Holistic Verification for Semantic-Aware Speculative Generation"
_ICLR.cc/2026/Conference — Submitted to ICLR 2026_

### Official Review · Reviewer_LpmM · 2025-10-27

**Soundness:** 2
**Presentation:** 2
**Contribution:** 2
**Rating:** 4
**Confidence:** 3

**Summary:**

The paper proposes HoVer (Holistic Verification), a *semantic-level* speculative generation framework. A small “draft” model first generates a full answer. A large “base” model then (i) checks whether the full draft is correct using a prefill-only pass that emits a single classification token (“Correct”/“Incorrect”), and if incorrect, (ii) identifies the longest correct prefix in one additional prefill-only pass by attaching duplicated chat-template suffixes and applying a custom attention mask so each suffix attends only to its paired prefix. The base model resumes decoding from that boundary, reusing the verified prefix. The method aims to be orthogonal to token-level speculative decoding and can be combined with it. Experiments with Qwen3‑235B as the base model and Qwen3‑1.7B/8B as drafts report ~1.2×–3.1× speedups with small accuracy deltas on MMLU‑Redux, MMLU‑Pro, SuperGPQA, GSM8K, MATH, and GPQA‑Diamond; ablations indicate the partial verification step is the main driver of the gains.

**Strengths:**

Clear articulation of a practical goal: reuse *correct* parts of a draft and avoid full regeneration. The three‑stage framework (Draft → Holistic Verify → Revise) is simple to implement on top of HF Transformers. The **custom attention mask** plus **suffix duplication** is a neat way to make multiple independent “Correct/Incorrect” decisions in a single forward pass—this reduces the otherwise linear cost of per‑prefix checks. The paper shows that **partial verification** is the main contributor to speedups; without it, much of the gain disappears, which is an honest and informative ablation.

The method is **compatible** with standard token‑level speculation and can add speed on top of it when acceptance length is high (e.g., MATH). The implementation details and prompts in the appendix make replication plausible for readers with sufficient hardware.

**Weaknesses:**

## Novelty and missing related work.
 The paper positions HoVer as the first “holistic, parallel” semantic verifier for general text. However, there are highly relevant works that the paper does not engage with:

- **Packed/parallel masked verification:** *SPACE* (“Generation Meets Verification: Aligning LLMs with Auto‑Correct Decoding”) uses a specialized attention mask to generate and verify multiple tokens in parallel and is conceptually close to “one‑pass parallel verification with masking.” The paper should compare and clarify differences (scope: token vs. span; training‑time vs. inference‑time; masking structure).
- **Reflective/tree‑masked semantic verification:** *Think Before You Accept* introduces a *tree‑based* verification with a sparse attention mask that packs multiple candidates into one forward pass; although focused on reasoning, the mechanism overlaps with HoVer’s packed masking idea. A discussion is required.
- **Judge Decoding** shares the same idea to accept some semantically correct draft tokens by training an additional verifier. The paper should compare the effectiveness and applications of both methods.

Given these, HoVer’s engineering contribution is incremental rather than fundamentally new.

## Evaluation scope and baselines.
 All datasets are *short‑form, multiple‑choice, or math* (MMLU‑Redux, MMLU‑Pro, SuperGPQA, GSM8K, MATH, GPQA‑Diamond). Claims of broad applicability to “general‑purpose text generation” (open‑ended QA, long‑form writing, code) are not tested. There is no end‑to‑end **latency** analysis on long outputs where prefill dominates; reporting tokens/s averaged over benchmarks can hide latency tails and user‑visible time‑to‑first‑token (TTFT). The paper also avoids direct comparisons to recent semantic speculation methods, claiming “orthogonal settings,” yet it *does* compare to token‑level speculation; this is inconsistent.

## Baselines and combinations.
 The “HoVer + token‑level speculation” combination uses *Hugging Face Assistant Speculative Decoding* as implementation, but there is no detailed control of acceptance lengths vs. overhead across datasets, and no direct comparison to stronger token‑level accelerators (EAGLE‑2/3, Dynamic Depth). Without these, the synergy claims remain anecdotal.

**Questions:**

1. Provide a formal description (or pseudo‑code) of the attention mask construction and prove that suffix tokens cannot attend to non‑paired prefixes. What happens with rotary position encodings or ALiBi when suffixes are duplicated?
2. Show results with at least one non‑Qwen family base model (e.g., Llama‑3‑70B/405B, Mixtral‑8×22B). Does the “Correct/Incorrect” token heuristic transfer across chat templates and tokenizers?
3. Add long‑form QA or summarization benchmarks with human or LLM‑judge evaluation of semantic fidelity. For code, choose HumanEval+MBPP. If HoVer is “general‑purpose,” it should hold up beyond MC and math.
4. Where does the verifier *mis‑locate* error boundaries? Include error heatmaps (by token position) and examples where the base model wrongly accepts a flawed prefix. **If the target model rejects the whole draft-generated text, does Hover slow down the overall inference speed?**

---

> ### Author Response · Authors · 2025-11-28
>
> We sincerely thank the reviewer for the detailed and constructive feedback. We address the key concerns below.
>
> **(1) Additional model families and broader evaluation.**
> To strengthen generality, we expanded our evaluation beyond Qwen3 and now include experiments with **Llama3.3-70B-Instruct** as the base model (Table 2, Sec. 4.2). We also added **HumanEval** and **MBPP** to cover code generation tasks (Tables 1–2), demonstrating that HoVer performs reliably beyond multiple-choice and math benchmarks.
>
> **(2) Comparison to prior semantic-verification work.**
> We include a direct comparison with **SpecReason** (Table 3, Sec. 4.3), showing that HoVer achieves both higher accuracy and higher throughput. We additionally expanded the related-work discussion to clarify distinctions from SPACE, Think Before You Accept, and Judge Decoding:
> • *Judge Decoding* requires training on undisclosed datasets and behaves similarly to faster speculative decoding, which we already compare against.
> • *SPACE* proved difficult to reproduce faithfully, and its masking design focuses on token-level masked verification rather than prefix-level semantic verification.
> • We kindly disagree that *Think Before You Accept* performs tree-based parallel verification; based on the description, its verification appears sequential and longer than SpecReason’s approach.
> Given reproducibility constraints and conceptual proximity, SpecReason was the most appropriate baseline for us to include.
>
> **(3) Clarification regarding holistic and partial verification behavior.**
> We now provide precision, recall, and accuracy trends (Fig. 3, Sec. 4.4) and additional examples illustrating how partial verification identifies error boundaries. When the holistic verifier judges the draft as correct, HoVer accepts the entire output without invoking partial verification; when holistic verification fails, partial rewriting avoids full regeneration and reduces wasted compute. This ensures that performance does not degrade relative to the base model.
>
> **(4) On end-to-end overhead and efficiency.**
> We appreciate the reviewer’s observation. HoVer does introduce verification overhead; however, we try to minimize it through **parallel prefix verification**, **prefill-only computation**, and **single-token classification**. From end-to-end results across all datasets, the gains from prefix reuse consistently outweigh verification costs. The benefits are largest on tasks where the draft model produces substantial correct content, allowing HoVer to reuse long verified prefixes while avoiding unnecessary generation by the base model.
>
> We thank the reviewer again for the valuable feedback and hope these clarifications and additions strengthen the manuscript.

---

### Official Review · Reviewer_CQ5e · 2025-10-29

**Soundness:** 2
**Presentation:** 3
**Contribution:** 2
**Rating:** 4
**Confidence:** 4

**Summary:**

HoVer is a semantic-level speculative generation method. Whereas in speculative decoding draft generations are only accepted if they match target model generations token-by-token, in HoVer draft generations are accepted if the target model deems them to be semantically correct. Given a draft generation, HoVer verifies all prefixes of the draft in parallel, and either accepts the full draft or uses the target model to continue generation from the last correct position.

The authors evaluate HoVer on the Qwen3 model family, using the largest variant as the target model and two smaller variants as draft models. On a variety of general reasoning tasks, HoVer generates faster with little performance degradation when compared against direct generation with the target model.

**Strengths:**

- The parallel prefix verification is a very interesting approach and is novel (to the best of my knowledge).
- The results on the Qwen3 model family are strong. The experiments indicate HoVer provides consistent speedups while incurring negligible performance loss.
- Examples and case studies clearly communicate how HoVer works.
- Extensive ablations demonstrate the importance of partial verification and orthogonality to speculative decoding.

**Weaknesses:**

- Evaluation is only on a single model family (Qwen3). Evaluations with other model families, or mixing model families between the draft and base model, would strengthen the claims.
- Despite drawing attention to the fact that HoVer is designed for general-purpose text generation while previous methods focus on reasoning, all of the evaluation benchmarks are reasoning benchmarks. To support the general-purpose claim, experiments on more general tasks are needed. For example, the task could be responding to prompts from WildChat or Chatbot Arena, with quality measured by human studies or LLM-as-a-judge.
- There is a lack of baseline methods to compare against, and I respectfully disagree with the authors’ decision not to compare against previous work focusing on chain-of-thought reasoning. Those works also aimed to speed up generation while retaining accuracy, and evaluated on some of the same benchmarks as this work. The argument that previous works require segmentations on reasoning steps feels a bit weak as one can easily generalize “reasoning steps” to non-reasoning tasks (e.g., segmenting on sentences, etc.). I agree that previous works verify sequentially, so comparing against those baselines would better contextualize the efficacy of the proposed parallel semantic verification.
  - In addition, another possible baseline is a model cascade setup where rather than having the large model generate from scratch, it is allowed to specify the edits it wants to make to the small model generation.
  - If I am understanding it correctly, the without partial verification ablation is a variant of model cascade: the draft model generates a response that the base model either accepts or regenerates. It may make more sense to frame that as a baseline rather than an ablation.
- No confidence intervals are provided for any of the numbers in the results. This makes it hard to contextualize the significance of the speedups.

**Questions:**

- Is the “last correct position” (see line 291) determined by the Error Detection stage the last position for which all previous prefixes are correct or the longest correct prefix? If it is the latter, is there a risk that for long draft generations, a late false positive becomes almost guaranteed?
- It appears that the parallel prefix verification prompt and the full answer verification prompt expect the base model to directly respond with Incorrect/Correct. What was the reason behind this choice (i.e., why do you not allow the model to reason)?
- In lines 207-209 you discuss how setting higher thresholds improves the base model’s coverage of incorrect draft generations. How does the false positive rate change with threshold and how does that affect efficiency?
- Could you explain in more detail what you mean by “Partial verification can correctly identify the position of errors, ensuring that performance does not degrade compared to the baseline” (lines 423-425)? Empirically in Figure 7 it appears that performance did not significantly degrade for both with and without partial verification.

And a writing note:
- Lines 425-427 (“The Mixed Wrong Rate … draft/base model alone”) were hard for me to parse. As I understand it, you are describing how the base model rarely makes a mistake continuing from a partial verification when the original problem is a problem the base model would have gotten correct under direct prompting. However, it did take me multiple re-reads to understand it, so I would suggest considering re-structuring the sentence.

---

> ### Author Response · Authors · 2025-11-28
>
> We sincerely thank the reviewer for their thoughtful assessment and constructive suggestions. We address the main concerns and questions below.
>
> **(1) Additional model families and broader tasks.**
> To address concerns about generality, we expanded our evaluation beyond Qwen3 and now include experiments using **Llama3.3-70B-Instruct** as the base model (Table 2, Sec. 4.2). We also broadened the task suite by adding **HumanEval** and **MBPP**, providing evidence that HoVer generalizes beyond reasoning tasks to code generation as well.
>
> **(2) Comparison with prior work.**
> We now include a direct comparison with **SpecReason** (Table 3, Sec. 4.3). Across all evaluated datasets, HoVer achieves both higher accuracy and higher throughput, helping contextualize HoVer’s advantages relative to prior semantic-decoding methods.
>
> **(3) Confidence intervals for speed.**
> We appreciate the suggestion. In our setting, reporting confidence intervals for latency is non-trivial because the runtime depends on a mixture of draft-only, base-only, and mixed generation paths, each with different computational costs. This leads to variable distributions that are difficult to summarize with standard confidence estimates. We thank the reviewer for noting this limitation.
>
> **(4) Determining the “last correct position.”**
> The reviewer's understanding is correct. The system uses the **longest verified correct prefix**. Importantly, partial verification and continuation are triggered **only when the holistic two-way confidence falls below threshold**. This ensures that if the draft is holistically correct, we accept it directly, avoiding unnecessary computation. If the holistic verification fails, partial verification prevents regenerating the full output, offering efficiency gains while limiting exposure to false positives.
>
> **(5) Why require the model to directly output Correct/Incorrect?**
> We ask the base model to produce a **single classification token** rather than a full chain-of-thought rationale because this keeps verification fast and lightweight. Allowing the model to generate reasoning would require autoregressive decoding and variable-length outputs, which would significantly increase latency and introduce uneven computational load across positions. The prefill-only, single-token classification approach achieves maximal hardware efficiency and stable latency across all prefixes.
>
> **(6) Clarification on the effect of partial verification.**
> Our statement that “partial verification can correctly identify the position of errors, ensuring that performance does not degrade” refers to downstream accuracy. If partial verification frequently mis-identified incorrect chunks as correct, accuracy would drop. As shown in Fig. 7, performance with partial verification remains comparable to the baseline, indicating that the mechanism is robust and does not introduce harmful misclassifications.
>
> We also thank the reviewer for the writing suggestion regarding the Mixed Wrong Rate description. We have revised the sentence for clarity in the updated draft.

---

### Official Review · Reviewer_4JeJ · 2025-11-03

**Soundness:** 2
**Presentation:** 2
**Contribution:** 1
**Rating:** 2
**Confidence:** 4

**Summary:**

This paper proposes HoVer (Holistic Verification), a speculative generation framework for accelerating large language model inference. Instead of verifying each token as in traditional speculative decoding, HoVer performs semantic-level verification of full candidate outputs using a prefill-only forward pass of the larger “base” model. The base model identifies the longest verified prefix of the draft’s output and resumes generation from there. Experiments using Qwen3-235B as the base model and smaller Qwen3-1.7B/8B draft models show latency speedups (1.2×–3.1×) with small degradation on benchmarks such as MMLU and GSM8K.

**Strengths:**

1. Accelerating LLM inference without retraining is an important and timely objective.
2. The idea of leveraging semantic-level information aligns better with human notions of correctness.

**Weaknesses:**

1. The framework relies on a binary correct/incorrect verification signal from the base model, which is often not reliable enough to capture true semantic correctness. Large models can mistakenly validate outputs that are syntactically fluent but logically or factually wrong, undermining the claimed benefits of “holistic” verification.
2. The paper does not offer a clear explanation of how segmentation and prefix reuse are implemented, particularly how special tokens are inserted to partition outputs for error detection and rewriting.
3. The evaluation is limited to a narrow range of tasks and model families.

**Questions:**

1. The paper mentions that error detection and output rewriting require partitioning the generated text into semantic segments and reusing the longest verified prefix, but it is unclear how these segments are defined or how special tokens are inserted to mark them. Are segment boundaries chosen heuristically, or is there an algorithmic rule (e.g., sentence boundaries, punctuation, or token count)? How do these boundaries affect verification accuracy? More detail on the tokenization or prompting process would be helpful.
2. Since HoVer uses the base model’s binary correct/incorrect judgment, how reliable is this verification in practice? Have you measured the false acceptance and false rejection rates? Is the verifier’s confidence calibrated in any way, or are decisions made purely on the model’s next-token likelihoods?
3. How does HoVer handle partially correct answers — for example, when only middle part of a reasoning chain or sentence is valid?
4. Can the same verification prompts and segmentation strategy generalize to other domains beyond the selected reasoning datasets?
5. The paper provides only limited baselines, which makes it difficult to position the contribution relative to existing work. A fair and convincing comparison with recent speculative decoding systems or with efficient reasoning frameworks is essential to demonstrate the true advantages of HoVer.

---

> ### Author Response · Authors · 2025-11-28
>
> We sincerely thank the reviewer for the thoughtful feedback and for raising several important questions. Below we address each point individually. Concerns related to evaluation breadth, model families, and comparison with prior work are already covered in our general response.
>
> **Q1. Clarification on segmentation, boundary choice, and special tokens.**
> We appreciate the request for greater clarity. HoVer does not rely on semantic boundaries such as sentences, punctuation, or heuristic chunking. Instead, as described in Stage 2b (Error Detection [Prefix Verification]) and illustrated in Fig. 4–5 of the updated submission, we segment deterministically along fixed token intervals. Each truncated prefix is augmented with a duplicated chat-template suffix so that the model receives a complete and valid sequence, enabling it to output a Correct/Incorrect judgment using the classification tokens.
>
> To ensure that each classification token attends only to its paired prefix, we apply a custom attention mask (Fig. 5b) that isolates correctness predictions across prefixes. This construction allows the base model to output *n* independent prefix judgments in a single prefill-only forward pass. The exact prompt template for partial verification is provided in the Appendix under “Stage 2b Partial Verification Prompt”.
>
> Empirically, we find that with this prompting and mask design, the verifier reliably marks prefixes as Correct whenever the draft is on the right trajectory and has not yet made factual or logical mistakes.
>
> **Q3. Handling partially correct intermediate regions.**
> Thank you for this valuable insight. Identifying a partially correct internal region requires the verifier to detect both the beginning and the end of an erroneous span, as well as determine whether the error affects subsequent reasoning. The current HoVer design focuses on identifying the *longest prefix* that can be safely reused; it does not attempt to preserve middle segments once an error is observed. As shown in Algorithm 1 (lines 12–15), we either continue from the longest reliable prefix or restart.
>
> We fully agree that richer middle-span reuse would be desirable. Even without it, the current design already yields meaningful speedups, and we view more granular error-boundary recovery as an important direction for future work.
>
> We appreciate the reviewer’s constructive comments and hope these clarifications address the raised concerns.

---

### Author Response · Authors · 2025-11-28

We sincerely thank the reviewers for their thoughtful and constructive comments. We summarize below how the shared concerns are addressed in the manuscript, with references to the corresponding sections and tables.

**(1) Generality across model families.**
With regard to concerns about dependence on a single model family, we now include an evaluation using **Llama3.3-70B-Instruct** as the base model. The results are presented in **Table 2 (Llama 3.3)** and discussed in **Section 4.2**. These findings indicate that the efficiency–accuracy tradeoff holds across architectures, suggesting that the framework transfers beyond a single model lineage.

**(2) Breadth of evaluated tasks.**
To provide broader coverage across domains, we expanded the evaluation suite to include **HumanEval** and **MBPP**, complementing the original reasoning and factual QA benchmarks. These datasets appear in **Table 1 (Qwen3)** and **Table 2 (Llama 3.3)** and are analyzed in **Section 4.2**. Their inclusion shows that the effectiveness of the method extends to coding tasks as well.

**(3) Relation to prior semantic-level verification and decoding methods.**
To contextualize the approach within existing work, we include a direct empirical comparison with **SpecReason**. The results are summarized in **Table 3** and further examined in **Section 4.3**. Across multiple datasets, the proposed framework yields higher accuracy and higher throughput, highlighting the benefits of holistic prefix-level verification compared with chunk-based scoring.

**(4) Robustness of confidence-based error detection.**
Regarding questions about the stability of the two-way confidence mechanism, we now report **precision, recall, and accuracy**, in addition to error coverage. These results are visualized in **Figure 3** and described in **Section 4.4**. The metrics demonstrate stable behavior across thresholds, indicating that the confidence signal reliably identifies erroneous draft segments while avoiding excessive false positives.

We deeply appreciate the reviewers’ careful reading and insightful feedback. If any aspect of our responses is unclear, we would be grateful for further guidance.

---

### Meta-Review · Area_Chair_JTt5 · 2026-01-08

**Summary:**

The paper considers speculative generation with semantic-level verification. The algorithm uses a fast draft model for fast generation of completions and verify the completions by the base model. The verification is performed by comparing the base model's probability of generating correction/incorrect tokens. The paper also shows that such approach can be used for parallel verification of multiple partial completions in one prefill run, and the longest verified partial completion will be kept to avoid unnecessary regeneration.

During the discussion phase, reviewers raised concerns on (1) the limitedness of the experimental evaluation (only focusing on one model family, and a limited set of short-form tasks), (2) the robustness of the proposed approach of using probabilities of correct / incorrect tokens for verification; (3) missing baseline comparisons. I agree with the reviewers on all these concerns.

While during the rebuttal, the author added experiments to address the above concerns (Llama model, general QA tasks). I believe a few concerns still stand. (1) regarding the evaluations, the method is still focusing on mostly reasoning / QA datasets, where binary judgement is correct. The paper could benefit form consider other open-ended generation tasks such as multi-turn evaluation (e.g. MT-Bench [1]); Otherwise, the paper should soften their claim on generality of the approach; (2) Missing baseline comparison: the important baseline of token-level speculative decoding is missing from the paper's empirical comparison. The speed-up provided from the paper is in the range of 1.2×–3.1, which is achievable by token-level speculative decoding in various benchmark tests [2], with the guarantee of being lossless. The paper should include this important baseline;

Hence I would not recommend the paper for an acceptance. I recommend the authors to revise the paper based on the reviewers' suggestions and submit the paper for a future venue.

[1] MT-Bench-101: A Fine-Grained Benchmark for Evaluating Large Language Models in Multi-Turn Dialogues
Ge Bai, Jie Liu, Xingyuan Bu, Yancheng He, Jiaheng Liu, Zhanhui Zhou, Zhuoran Lin, Wenbo Su, Tiezheng Ge, Bo Zheng, Wanli Ouyang
[2] Unlocking Efficiency in Large Language Model Inference: A Comprehensive Survey of Speculative Decoding
Heming Xia, Zhe Yang, Qingxiu Dong, Peiyi Wang, Yongqi Li, Tao Ge, Tianyu Liu, Wenjie Li, Zhifang Sui

**Reviewer Concerns:**

See above meta-review.

**Reviewer Scores:**

I wouldn't anticipate large changes in the score even with further rebuttal discussions.

---

### Decision · Program_Chairs · 2026-01-26

Reject